# The Roles of Ubiquitin-Binding Protein Shuttles in the Degradative Fate of Ubiquitinated Proteins in the Ubiquitin-Proteasome System and Autophagy

**DOI:** 10.3390/cells8010040

**Published:** 2019-01-10

**Authors:** Katarzyna Zientara-Rytter, Suresh Subramani

**Affiliations:** Section of Molecular Biology, Division of Biological Sciences, University California, San Diego, CA 92093-0322, USA; ssubramani@ucsd.edu

**Keywords:** ubiquitin-proteasome system, selective autophagy, proteostasis, protein quality control, ubiquitination

## Abstract

The ubiquitin-proteasome system (UPS) and autophagy are the two major intracellular protein quality control (PQC) pathways that are responsible for cellular proteostasis (homeostasis of the proteome) by ensuring the timely degradation of misfolded, damaged, and unwanted proteins. Ubiquitination serves as the degradation signal in both these systems, but substrates are precisely targeted to one or the other pathway. Determining how and when cells target specific proteins to these two alternative PQC pathways and control the crosstalk between them are topics of considerable interest. The ubiquitin (Ub) recognition code based on the type of Ub-linked chains on substrate proteins was believed to play a pivotal role in this process, but an increasing body of evidence indicates that the PQC pathway choice is also made based on other criteria. These include the oligomeric state of the Ub-binding protein shuttles, their conformation, protein modifications, and the presence of motifs that interact with ATG8/LC3/GABARAP (autophagy-related protein 8/microtubule-associated protein 1A/1B-light chain 3/GABA type A receptor-associated protein) protein family members. In this review, we summarize the current knowledge regarding the Ub recognition code that is bound by Ub-binding proteasomal and autophagic receptors. We also discuss how cells can modify substrate fate by modulating the structure, conformation, and physical properties of these receptors to affect their shuttling between both degradation pathways.

## 1. Introduction—Protein Quality Control Mechanisms

At the cellular level, cells need to recycle old, abnormal, or dysfunctional proteins and maintain a “healthy” balance between newly-synthesized proteins and existing ones in order to maintain the efficient functioning of cellular pathways and systems. Approximately one-third of newly-synthesized proteins in human cells require removal due to their improper folding, despite the presence of chaperones that prevent protein aggregation and assist in protein folding [1]. Because homeostasis of the cellular proteome (proteostasis) ensures successful cell development, health aging, stress tolerance, and adaptation to changing environmental conditions or even resistance to infections by pathogens, any imbalance in protein homeostasis (triggered, for example, by accumulation of misfolded proteins) is widely accepted to cause diseases [2]. Because of the fundamental importance of proteostasis, cells have evolved a sophisticated housecleaning protein quality control (PQC) system, which is based primarily on two major degradation pathways known as the ubiquitin (Ub)-proteasome system (UPS) and selective macroautophagy (hereafter, autophagy) [2,3].

In the UPS, substrates for degradation are firstly ubiquitinated via an enzymatic cascade involving E1 Ub-activating enzyme(s), E2 Ub-conjugating enzyme(s), and E3 Ub ligases [4]. Subsequently, most of these ubiquitinated (generally poly-ubiquitinated) proteins are recognized and turned over by the proteasome, which is a large multi-subunit complex of proteases and regulatory proteins (for more details about the UPS and proteasome structure, see References [2,5,6,7]). The UPS is responsible for the degradation of most cellular soluble proteins [2]. Proteins targeted to the UPS are individual misfolded or damaged proteins (housecleaning degradation), as well as unneeded or redundant proteins that function in regulatory processes (regulatory degradation). Therefore, there is no doubt that the UPS plays a pivotal role in PQC. However, if ubiquitinated substrates are resistant to proteasomal degradation, or when the UPS is overwhelmed or impaired, these proteins may become aggregation-prone and may form aberrant aggregates that become inaccessible to proteasomal proteases [8,9] because of the intrinsic size limitation of proteins subject to degradation by the UPS. In such cases, to maintain proteostasis, these protein aggregates will be immediately redirected to autophagy, or temporarily stored in aggresomes-perinuclear compartments with high autophagic activity located at the microtubule organizing center (MTOC)- to reduce their toxicity [10,11]. Based on recent studies in mammalian cells, such initial accumulation of misfolded proteins in aggresomes per se is tolerable. However, the absence of replenishment of amino acids and Ub molecules caused by the longer sequestration of many ubiquitinated proteins could become highly dangerous for cell viability. Therefore, as a consequence, an alternative pathway, autophagy, is induced [12,13]. Thus, regardless of whether a protein is degraded instantly or is stored in the aggresome initially, the fate of poly-ubiquitinated proteins destined for degradation is pre-ordained in the sense that, when proteasomal clearance is inhibited, these proteins will be degraded at some point via autophagy, which is a major pathway for degradation of large substrates, such as aggregates and organelles [2].

Autophagy is a catabolic pathway conserved from yeast to mammals and plants. It involves the conserved action of more than 20 specific AuTophaGy related (ATG) proteins that comprise the core machinery, which mediates the formation of a double-membrane structure called the autophagosome. This structure is decorated by ATG8/LC3/GABARAP (autophagy-related protein 8/microtubule-associated protein 1A/1B-light chain 3/GABA type A receptor-associated protein) protein family members. The ATG8/LC3/GABARAP proteins are Ub-like proteins that share structural features with Ub, such as two amino-terminal α helices and a Ub-like core [14,15,16,17,18,19,20]. During autophagosome formation, these proteins undergo a unique Ub-like conjugation to phosphatidylethanolamine on the isolation membrane (phagophore, pre-autophagosomal structure), which expands around the substrate(s) and encloses it in a vesicle (autophagosome) for further delivery to, and clearance within, the vacuole (in yeast and plants) or lysosome (in mammals). ATG8/LC3/GABARAPs are proteins associated with mature autophagosome and, therefore, serve as bona fide markers for this unique vesicle (for more details about the autophagic process and the machinery, see References [21,22,23]).

Autophagy in eukaryotes was initially considered to be a non-selective, bulk system, but specific receptors precisely recognize and deliver cellular substrates (proteins or organelles) exclusively to autophagosomes and then the vacuole/lysosomes [24]. Aggregates of misfolded proteins are among the various substrates that can be selectively targeted for autophagic clearance [25,26,27,28,29,30,31,32,33]. In this process, called aggrephagy, particular autophagic cargo receptors recognize ubiquitinated aggregates and link them to the expanding isolation membrane through their interaction with ATG8/LC3/GABARAP protein family members for subsequent degradation [25].

Based on several differences in mechanisms of action, the UPS and autophagy were initially viewed as two mutually-independent degradative systems in PQC, with distinct receptors and adaptors. However, in view of the fact that Ub and the ubiquitination process (which refers to the conjugation of the Ub to lysine residues, or rarely cysteine, of the target protein destined for degradation) [34,35,36,37] are used by both the UPS and the autophagy pathways, there is increasing evidence that they should not be studied separately because of the dynamic interplay between these two protein degradation systems [38,39,40,41,42,43].

In this review, we will summarize the current knowledge regarding the Ub code (the varying patterns of ubiquitination of a target protein) in the UPS and autophagy contexts. We will also discuss how cells make decisions and modify the fates of substrates based on the structure and properties of the Ub-binding protein shuttles that are used for substrate degradation.

## 2. Ub and Ubiquitination at the Crossroads of the UPS and Autophagy

Both the UPS and autophagy pathways use ubiquitination as a labeling system for substrate recognition. Ub is a highly-stable, 76 amino acid protein, which is almost identical in all eukaryotes [44]. Such high evolutionary conservation points to the importance of Ub and its recognition code for proteostasis and suggests enormous pressure to preserve the structure of this protein since many of its surfaces are recognized by Ub-binding domains (UBDs) in proteins [44]. Through the coordinated activity of the E1, E2, and E3 enzymes mentioned earlier, Ub is covalently attached via its C-terminal glycine to usually one or more lysine (and rarely cysteine) residues of the target protein. Attachment of just one Ub is named mono-ubiquitination and is a common mechanism of protein post-translational modification that regulates protein function and trafficking (Figure 1). Multiple Ubs may also become linked to several independent lysine residues in one protein causing multi-mono-ubiquitination. Alternatively, through the addition of other Ubs to a Ub previously conjugated to a substrate at a lysine residue, the target may be poly-ubiquitinated (Figure 1).

Based on the number of Ub molecules attached to each other, short (starting with only two Ub moieties) or long (more than 10 Ubs) chains can be formed. Additionally, since poly-ubiquitination can occur via bond formation on Met1 or any Lys residue within Ub (e.g., Lys6, Lys11, Lys27, Lys29, Lys33, Lys48, Lys63), which, by the way, are well-distributed on the surface and oriented in distinct directions, different types of homogeneous chains (e.g., Lys11-linked, Lys48-linked, or Lys63-linked Ub chains), or chains with mixed topology, can be generated (Figure 1) (for more information about the ubiquitination process and Ub chains, see References [44,45,46]). Such flexibility of the ubiquitination process (from mono- and multi-mono- to poly-ubiquitination) and structural diversity of the Ub chains are commonly accepted as the foundation of the “Ub recognition code” that potentially encodes information about protein fates in the cell. Nuclear magnetic resonance (NMR) and crystal structure analyses of various Ub chains revealed that different chain types adopt distinct conformations with either a so-called “compact” one, where neighboring Ub molecules in close proximity interact with each other and form a higher-order assembly or “open” structures, where linked Ub moieties have greater conformational freedom because they are too far from each other to form tight interactions [44]. Contrary to Met1-linked and Lys63-linked Ub chains that adopt “open” conformations, the Lys6-linked, Lys11-linked, and Lys48-linked Ub chains display “compact” conformations (Figure 1) [44]. Therefore, it is commonly accepted that Ub-binding proteins might recognize and preferentially bind to proteins with particular ubiquitination status, or with specific Ub chain types by, for example, sensing the distance between Ub molecules or the relative orientation of Ub surfaces within a chain [44].

Thus, it is not surprising that many Ub recognition receptors with diverse UBDs have been described. At least 21 different classes of Ub-binding modules have been characterized so far [47]. Among them, the most interesting for the UPS and autophagy degradation systems are: (1) the Ub-associated (UBA) domain present in proteasomal (Rad23, Dsk2, Ddi1) and autophagic receptors (sequestosome 1 (SQSTM1), also known as p62, and neighbor of BRCA1 gene 1 (NBR1) in mammals or Joka2/NBR1 in plants), (2) the coupling of Ub to ER degradation (CUE) domain present in Cue5 protein in yeast and toll-interacting protein (TOLLIP) in mammals, (3) the Ub-interacting motif (UIM) present in the proteasomal protein S5a/Rpn10/Pus1 (in higher eukaryotes, *S. cerevisiae*, and *S. pombe*, respectively), (4) zinc fingers such as the zinc-finger Ub-binding domain (ZnF UBP) found in histone deacetylase 6 (HDAC6) or the Ub-binding zinc finger (UBZ) present in the autophagic receptor, nuclear dot protein 52 kDa (NDP52), also known as calcium-binding and coiled-coil domain-containing protein 2 (CALCOCO2) (in mammals), (5) UBD in A20-binding inhibitor of NF-κB (ABIN) proteins and NF-κB essential modulator (NEMO) (UBAN) domain located in another mammalian autophagic receptor, optineurin (OPTN), (6) and a pleckstrin-like receptor for Ub (PRU) domain, present in the Ub receptor at the proteasome, Rpn13.

We present below a brief description of the Ub-binding modules present in several shuttling receptors involved in the UPS and in autophagy.

## 3. Ub-Binding Proteins Functioning in the UPS

Among the Ub-binding receptors required for the proteasomal clearance of target proteins are those that are part of proteasome itself and others that only temporarily associate with it during substrate delivery. The intrinsic proteasomal (proteasome-associated) Ub receptors, Rpn10 and Rpn13, can only uptake poly-ubiquitinated substrates that are located in a close proximity to proteasomes [48]. In contrast, shuttling Ub-binding receptors, which are not integral subunits of the proteasome, can recognize ubiquitinated cargos located far from the proteasome and bring them for degradation by the proteasomal clearing machinery [49,50,51,52,53,54,55,56,57,58,59]. Interestingly, known classical proteasomal shuttle proteins [60] contain an N-terminal, Ub-like domain (UbL) and a C-terminal Ub-associated domain/s (UBA), which enable them to deliver poly-ubiquitinated proteins to the proteasome for degradation.

The UbL domain is approximately 80 amino acids long and adopts a typical Ub fold, even though the sequence homology with Ub is limited [61]. The UBA domain is even smaller (approximately 40 amino acids) and adopts a specific fold composed of a bundle of three tightly packed α helices separated by two small flexible regions [62,63,64] in which the first and third helices create the major contact site for Ub binding. Yeast Rad23, Dsk2, Ddi1 proteins, and their homologues belong to the UbL-UBA protein family and are known as proteasomal shuttles [60]. Their domain architectures, together with those of known Ub-binding autophagic shuttling receptors [24,65,66,67], are presented in Figure 2.

### 3.1. Rad23

Rad23 was the first UbL-containing protein discovered in yeast [70] and has been the most extensively studied member of the UbL-UBA shuttle family. It is a medium-sized, multi-domain protein containing an N-terminal UbL and two UBAs (central UBA1 and C-terminal UBA2) (Figure 2). Due to various binding modules, Rad23 is considered a scaffold protein since it is involved in a large number of cellular processes including nucleotide excision repair (NER) and, of course, proteasomal degradation of poly-ubiquitinated proteins [71]. The N-terminal UbL domain of Rad23 binds to Rpn1 or Rpn10 of the proteasomal 19S regulatory particle [72,73]. Binding to Rpn1 or Rpn10 positions Rad23 at the epicenter of the base, in close proximity to the entrance of the proteasomal core particle [74]. Similarly, the human homologues of Rad23, HR23A and HR23B, also bind to the proteasomal 19S subunit to present poly-ubiquitinated substrates by interacting with S5a (the human homologue of yeast Rpn10) [75,76]. These proteasome-interacting proteins bridge poly-ubiquitinated proteins and the proteasome by binding Ub-conjugates through their UBA domains which, interestingly, bind mono-Ub and poly-Ub but with district specificities [77]. Based on Rad23 UBA domains studies, UBA1 recognizes Lys63-linked Ub chains and has a major impact on Ub binding to Rad23. In contrast, UBA2 prefers Lys48-linked Ub chains, but affects Ub binding by Rad23 only in a minor manner because deletion of the UBA2 domain does not strongly affect Rad23 binding of poly-ubiquitinated proteins [77,78,79,80,81].

Rad23 also stabilizes proteasome substrates and inhibits the formation of Lys48-linked Ub chains [79,82,83]. Rhp23, the *S. pombe* homologue of Rad23, protects poly-ubiquitinated conjugates against deubiquitination by the enzyme Ub-specific protease Y (UBPY), which disassembles tetra-Ub chains via its UBA domain. Similarly, Rad23 reduces Rad4 multi-ubiquitination and stabilizes Rad4 levels. Thus, Rad23 and its homologues appear to function both in degradation and stabilization of ubiquitinated substrates.

### 3.2. Dsk2 and Ubiquilin Protein Families

Dsk2, which was initially identified as a protein involved in spindle pole body duplication [84], is another member of the UbL-UBA protein family characterized by an N-terminal UbL domain and a C-terminal UBA domain (Figure 2). However, its N-terminal UbL domain weakly interacts with the proteasome and binds with lower affinity than the interaction of Rad23 with the UIM domain of Rpn10 [56,72,85,86,87,88,89]. Dsk2 association with proteasomes has been only found in the absence of Rpn10 because free Rpn10 competes for Dsk2 binding to the proteasome [89]. In addition, the binding specificity of the Dsk2 UBA domain is quite unclear. Previous reports noted that Dsk2 binds through its UBA domain to poly-Ub conjugates with a preference for Lys48-linked Ub [87]. However, another group has proven recently that Dsk2 has no substantial preference toward Lys48-linked Ub and binds both Lys48-linked and Lys63-linked, purified Ub chains, as well as to mono-Ub [90]. Contradictory results obtained by these two groups could be related to usage of different experimental approaches. In the first approach, a series of lysine mutants of Ub (e.g., K29R, K48R, K63R) were made and the extracts of yeast cells were used, which could include additional factors affecting Dsk2 binding predisposition to poly-ubiquitinated substrates. The second study used free Lys48- and Lys63-linked poly-Ub chains to measure the binding affinities of isolated Dsk2 UBA (amino acids 300-373) to mono- and di-Ub chains. In addition to its unclear mechanism of action, Dsk2 protects poly-ubiquitinated conjugates against deubiquitination, as seen in Rad23, preventing the disassembly of tetra-Ub chains, and stabilizing Ub conjugates [87,89].

Mammalian ubiquilin proteins are the closest homologues to yeast Dsk2 protein. The human genome contains four ubiquilins (UBQLN1–4). While ubiquilin1 is ubiquitously expressed and ubiquilin2 and ubiquilin4 are expressed in most tissues, ubiquilin3 is only expressed in testis. All ubiquilins are highly conserved and share high degrees of sequence and domain structural homology to each other. Like yeast Dsk2, all ubiquilins harbor UbL domains at their N-termini and UBA domains at their C-termini. Located between these two domains are STI1 (stress-inducible heat shock chaperonin-binding motif) motifs, which may confer ubiquilins with chaperone-like functions (Figure 2) [91,92]. The best characterized member of the this family is ubiquilin1, which is also known as PLIC-1 (protein linking integrin associated protein with cytoskeleton 1). This cytosolic protein not only delivers poly-ubiquitinated proteins to the 26S proteasome [56,93] but also associates with aggregates or aggresomes [94] and delivers substrates to lysosomes for degradation via autophagy and chaperone-mediated autophagy (CMA) [95,96]. Additionally, its involvement in the ER-associated degradation (ERAD) pathway has also been documented [97]. Therefore, it is not a surprise that the UBA domain of ubiquilin1 shows similar binding affinity to Lys48-linked and Lys63-linked Ub chains [77].

### 3.3. Ddi1-Like Proteins

DNA Damage-Inducible 1 (Ddi1) protein also belongs to a family of shuttle proteins targeting poly-ubiquitinated cargo for proteasomal clearance, but differs from the other shuttling proteins in its proteolytic roles and its interacting partner. Like Rad23 and Dsk2, it usually contains an N-terminal UbL as well as a C-terminal UBA domain, which has been lost in mammals (but not in plants) (Figure 2).

It also has additional unique domains, such as conserved retroviral protease (RVP) fold domain, followed by a putative protein rich in Pro, Glu, Ser and Thr (PEST) region and Sso-binding domains (Sso-BD) [68,69]. Besides transferring ubiquitinated substrates to the proteasome for degradation [55,58,98,99], Ddi1 participates in many other cellular processes [68,81,98,100]. While the Ddi1 UBA domain forms a characteristic UBA-Ub complex, the Ddi1 UbL domain has unusual binding preferences, which may explain why, in higher eukaryotes, the UBA domain has been lost. In contrast to the Ub and UbL domains of Rad23 and Dsk2, the UbL domain of Ddi1 does not interact with UBA domains or the UIM of Rpn10 [101] and associates very weakly with Rpn1 [102]. Instead it binds Ub quite strongly by forming a unique interface mediated by hydrophobic contacts and by salt-bridges between oppositely-charged residues of the Ddi1 UbL domain and Ub [103].

Because the UbL domain of Ddi1 binds both Ub and the proteasome, it has a different mechanism of docking to substrates in comparison with Rad23 and Dsk2 proteins and may not act as a classic shuttle receptor. Instead, Ddi1 might assist Rad23 and Dsk2 rather than compete with them for targeting ubiquitinated substrates to the proteasome. For example, via the sequential dissociation of UbL from ubiquitinated substrates and its association with Rpn1, Ddi1 could increase the concentration of Ub conjugates in the proximity of the proteasomal 19S subunits and, therefore, expose poly-ubiquitinated proteins to the proteasome-associated Ub receptors, Rpn10 and Rpn13. Alternatively, via hetero-dimerization between Ddi1 and Rad23 through their UBA domains, a tandem (Ddi1-Rad23) shuttle could be formed [104]. Moreover, dimerization of Ddi1 through its RVP domain, which has been documented, might allow its simultaneous binding to an ubiquitinated substrate and to the proteasomal 19S subunit [103].

Loss of the UBA domain in the human Ddi1-like protein, DDI2, is partially restored by the presence of a novel UIM-like motif at the C-terminus of DDI2, which weakly but specifically binds mono-Ub [105]. Such a motif was not identified in human DDI1 [105].

## 4. Ub-Binding Receptors in Autophagy

Another group of shuttles plays a key role in both the recognition and the delivery of cytoplasmic substrates for autophagic clearance. In mammals, Ub tags serve as a general mechanism for labeling various substrates (from individual proteins through organelles to pathogens) for autophagic turnover. In yeast, recognition of autophagic cargo is not mediated by a common tag, and Cue5 is the only yeast receptor for Ub-dependent autophagy [106]. In contrast, in mammals, several autophagic receptors associate with poly-ubiquitinated proteins to deliver these substrates for autophagic clearance. These include TOLLIP, the mammalian homologue of Cue5 [106], SQSTM1, and NBR1, which are the best characterized autophagic receptors that recognize a broad range of substrates and are involved in almost every type of selective autophagy [27,29,30,31,107,108,109,110,111,112,113], OPTN, which is involved in the autophagic clearance of ubiquitinated aggregates, pathogens, and mitochondria [33,114,115,116], and, finally, NDP52, which is required for the autophagic clearance of ubiquitinated mitochondria and pathogens [117,118,119].

Unfortunately, little is known about autophagic receptors in general and about Ub-dependent receptors in plants. So far, based on domain architecture homology, AtNBR1/NtJoka2, hybrid homologues of human SQSTM1 and NBR1 have been described [65,66] (Figure 2).

### 4.1. CUET Proteins

CUE-domain targeting (Cue5-TOLLIP; CUET) proteins are probably ancient autophagic receptors involved in the clearance of cytotoxic protein aggregates [106]. CUETs are functionally conserved from yeast to mammals. This group of proteins is represented by yeast Cue5 and its human homologue, TOLLIP, and binds Ub via its CUE domain [106,120]. However, the domain organizations in TOLLIP and Cue5 are notably different (Figure 2). TOLLIP contains, in addition to a CUE domain, a large N-terminal extension harboring a TOM1 (target of myb1 homologue)-binding domain and a phospholipid-binding Ca+^2^-dependent membrane-targeting module (C2), which are postulated to be required for other non-autophagic functions [121,122]. TOLLIP plays several roles within the cells such as in protein traffic by the endocytotic pathway and in Toll-like receptor (TLR)-mediated innate immunity responses [122,123]. Also, in contrast to Cue5, which contains one C-terminally located ATG8-interacting motif (AIM), TOLLIP contains two functional LC3-interacting regions (LIRs) located within the C2 domain [106].

Interestingly, the CUE domains of both Cue5 and TOLLIP have no preferences toward Lys48-linked or Lys63-linked Ub chains and, therefore, might not distinguish between various types of Ub conjugates [106]. Because the CUE domains of Cue5 and TOLLIP bind mono-Ub and poly-Ub chains of various types, it is suggested that CUET receptors clear cells of protein aggregates [124,125], which contain diverse Ub linkages as a consequence of clustering various proteins ubiquitinated by different E3 ligases. In agreement with such a hypothesis, TOLLIP binds Ub-conjugated proteins better than does SQSTM1. The CUE domain of TOLLIP binds free Ub with similar affinity as does SQSTM1, but its binding to Lys48-linked and Lys63-linked poly-Ub chains is stronger than that of SQSTM1, especially those of a longer chain length [106]. In view of this ability to bind Ub, it is surprising that TOLLIP is recruited to aggregates independently of its recognition of Ub [106]. Further studies on TOLLIP might reveal the mechanism involved in TOLLIP targeting to aggregates.

Additionally, with regard to TOLLIP’s Ub-binding propensity, two contradictory results were obtained. In contrast to a previous report [120], a recent study found no evidence for Ub-binding activity for the C2 domain and has proven only the involvement of the CUE domain in Ub recruitment, as it is in Cue5 [106]. However, constructs of different length, containing the C2 domain, were used in these two experiments. While Mitra et al. [120] clearly showed that the isolated C2 domain of TOLLIP (amino acids 54–182) bound Ub with a similar affinity as other known Ub-binding domains [53]. Lu et al. [106] used the N-terminal (amino acids 1-180) region of TOLLIP containing the C2 domain. It is possible that the N-terminal region of TOLLIP, which precedes the C2 domain, has an inhibitory effect on Ub binding to the C2 region. This matter also needs further clarification.

### 4.2. SQSTM1

SQSTM1 was the first selective autophagy receptor discovered in mammals and is also the best characterized one [108,126,127]. It is a multi-functional protein consisting of an N-terminal Phox-BEM1 (PB1) domain, which is a central ZZ-type zinc finger domain, and a C-terminally located UBA domain. SQSTM1 also contains one LIR and a KEAP1 (Kelch-like ECH-associated protein 1)-interacting region (KIR), as well as a functional nuclear localization signal (NLS) and a nuclear export motif (NES) (Figure 2) [108,127,128,129,130]. Although SQSTM1 does not have a specific UbL region seen in the proteins of the UbL-UBA family, it does have a PB1 domain resembling the UbL domain. This fact, in combination with the presence of a UBA region in SQSTM1, may explain some of the properties that SQSTM1 shares with other UbL-UBA proteins. For example, it interacts via its UBA domain with poly-ubiquitinated proteins with preference for the Lys63-linked Ub chains and binds to the proteasome via its PB1 domain, which, like the UbL domain, forms a Ub-like, β-grasp fold, recognizable by the proteasomal components, S5a and Rpt1 [131,132]. The presence of both the NLS and NES motifs allows SQSTM1 to shuttle between the nucleus and cytoplasm [128]. This protein is implicated in autophagy and in several other pathways including DNA repair [133], the KEAP1-NRF2 signal transduction pathway [134], and in Wnt [135] or NFκB signaling [136]. However, SQSTM1 is mainly known as an autophagic receptor [27,29,31,108,109,134,137,138]. The PB1 domain of SQSTM1 has a high potential for homo-oligomerization, as well as for hetero-oligomerization with PB1-containing proteins, such as NBR1 [29]. The interaction of SQSTM1 and NBR1 organizes SQSTM1 into higher oligomers, which promotes SQSTM1 clustering with Ub-positive aggregates during aggrephagy, wherein its functional LIR motif strongly associates with mammalian LC3/GABARAP [108,139].

In addition to its role as an autophagic receptor, SQSTM1 also regulates the accumulation and autophagic clearance of protein aggregates. Recent studies show that SQSTM1 co-localizes with HDAC6 at ubiquitinated aggregates and by binding to the catalytic deacetylase domain 2 (DD2) of HDAC6, SQSTM1 negatively regulates HDAC6 deacetylase activity, promotes F-actin network assembly at the MTOC, and assists autophagosome-lysosome fusion [140].

### 4.3. NBR1

Like SQSTM1, NBR1 contains a PB1 domain at its N-terminus, which is followed by the ZZ domain and an UBA domain at its C-terminus. However, instead of the single LIR motif in SQSTM1, NBR1 contains two LIRs (one conforming to the typical consensus, W/F/YXXL/I/V, and the other being atypical) [30]. Besides these modules, NBR1 also contains two coiled-coil (CC) domains involved in NBR1 homo-oligomerization. This is a feature that allows its association with the microtubule network through microtubule-associated protein 1B (MAP1B) [141]. Additionally, NBR1 also has, preceding the UBA domain, a JUBA (juxta-UBA) α-helix that binds lipids [142]. The in silico analysis of the NBR1 protein sequence reveals two potential NESs, but no NLS (Figure 2).

Both NBR1 and SQSTM1 play critical roles in the recruitment of aggregate-prone proteins, long-lived proteins, or damaged organelles into autophagosomes and in the formation of protein aggregates. However, the occurrence of several distinct, dissimilar modules in NBR1 and SQSTM1 also confers on these proteins some additional unique properties. While SQSTM1 is considered as a main receptor for aggrephagy, and NBR1 significantly contributes, it is not essential [27,31]. The unique JUBA region that allows NBR1 to anchor directly to organelle membranes makes it a necessary and sufficient receptor for pexophagy (peroxisome degradation via autophagy) [111,142]. Moreover, the PB1 and UBA domains of NBR1 differ from those of SQSTM1 in oligomerization ability and interaction with Ub, respectively. The PB1 domain of SQSTM1 has the ability to polymerize [143,144,145] while this domain of NBR1 is unable to homo-oligomerize. Yet, it can still bind another PB1 domain (such as that in SQSTM1) [143,146]. Recently, the UBA domain of NBR1 was described to have a much higher affinity for Ub than that of SQSTM1 [147].

### 4.4. Plant NBR1-Like Proteins

NBR1 from *Arabidopsis* [66] and Joka2 from tobacco [65] behave as hybrid proteins that combine features of mammalian SQSTM1 and NBR1 and contain a similar PB1, ZZ, and UBA domain architecture [65,66,148]. However, instead of one C-terminally located UBA, plant receptors contain two, non-identical UBA domains, where only the second domain (UBA2) binds Ub in vitro [66] (Figure 2). Plant NBR1-like proteins target insoluble, detergent-resistant, poly-ubiquitinated protein aggregates accumulated during plant abiotic stresses and non-assembled particles of a cauliflower mosaic virus (CaMV) for their degradation via NBR1-mediated autophagy [149,150].

Both AtNBR1 and NtJoka2 are strongly aggregating proteins [65,66,148,149]. Like SQSTM1, but in contrast to mammalian NBR1, each of them possesses an acidic/basic PB1 surface that allows these NBR1-like proteins to self-oligomerize through their PB1 domain [66,148]. Thus, no CC regions analogous to those in mammalian NBR1 are necessary for oligomerization in the plant NBR1-like receptors. Consistent with the mechanistic data regarding cluster formation by SQSTM1, the PB1 domain in Joka2 is sufficient for its oligomerization, but the C-terminal region containing both UBA1 and UBA2 domains additionally promotes this process [148]. In addition, like all known autophagic receptors, plant NBR1-like proteins directly bind to ATG8 protein family members. Several potential AIMs are present within their sequences and at least one of them, located between the UBA domains, is functional [66].

Moreover, based on studies with Joka2, these proteins possess a functional NES and an uncharacterized NLS, which allows shuttling between the nucleus and the cytoplasm, analogous to SQSTM1 [148]. The role of NBR1-like proteins in the nucleus can only be hypothesized at present based on plausible functional conservation between plant NBR1-like proteins and SQSTM1. It is unclear whether these proteins like mammalian NBR1 contain a functional JUBA domain.

### 4.5. OPTN

Optineurin (OPTN) is another multi-domain protein that interacts with many partners and has versatile functions beside its role as an Ub-dependent autophagic receptor. Among its interactors are proteins involved in membrane and vesicle trafficking (Rab8, huntingtin, myosin VI), cellular morphogenesis, NF-κB regulation, signal transduction, transcription regulation, and cellular division control [151,152,153,154,155,156]. OPTN regulates autophagic degradation of mitochondria and cytosolic *Salmonella enterica* in processes called mitophagy and xenophagy, respectively, as well as of protein aggregates in both a Ub-dependent and Ub-independent manner [32,157].

OPTN involvement in autophagy depends on the activation of its LIR motif via the phosphorylation of a Ser residue preceding the core sequence, W/F/YXXL/I/V, by the TANK-binding kinase 1 (TBK1) [114,116]. OPTN interacts with linear and Lys63-linked poly-Ub chains via its UBAN domain [158] (Figure 2). However, OPTN can recognize protein aggregates not only in a Ub-dependent manner through its UBAN domain, which is required to recognize ubiquitinated *Salmonella enterica* [116] or mitochondria [114,115], but also through its C-terminal CC domain in a Ub-independent manner [32]. Moreover, a recent report has established that HECT domain and ankyrin repeat-containing E3 ubiquitin protein ligase 1 (HACE1)-mediated Lys27-linked and Lys48-linked poly-ubiquitylation of Lys193 in OPTN might activate autophagic removal of oxidative damaged proteins by promoting the formation of a SQSTM1-OPTN complex [159]. These data are consistent with the observation that OPTN can positively modulate protein aggregation [32].

In addition, to these roles in autophagy, OPTN also drives autophagosome maturation by interacting with myosin VI (MYO6), which is involved in the fusion of autophagosomes with endosomes [160].

Furthermore, it must be stressed that OPTN, like other autophagic receptors discussed in this scenario, can oligomerize to form 420 kDa homo-oligomers, which, based on the 67 kDa monomer size, were judged to be hexamers [161]. Unlike other autophagic receptors, which are degraded by autophagy, OPTN is normally degraded by the proteasome pathway [162].

## 5. PQC Pathway Choice Based on the Ub Code

Poly-Ub chains consisting of Lys48-linkages are synthesized by many E3 ligases and, therefore, are among the most abundant in the cell. Their level rapidly increases when proteasomes are inhibited, which points to their involvement in proteasomal degradation [163,164,165,166]. Moreover, a variety of E3 ligases, including UBR box N-recogins [167,168], gp78 - also known as AMFR (autocrine motility factor receptor) [169], E6AP (E6-associated protein) [170], and SCF (stem cell factor) [171], assemble Lys48-linked poly-Ub chains for proteasomal degradation, while others like Parkin [172], TRIM13 (tripartite motif-containing protein 13) [173], and CHIP (carboxyl terminus of Hsc70 interacting protein) [174] assemble Lys63-linked Ub chains on substrates for autophagic turnover. Therefore, in the Ub recognition code, it is well established that Lys48-linked Ub chains typically mark a protein for proteasomal degradation, while mono-ubiquitination and Lys63-linked Ub chains seem to play a role in autophagy [175,176]. Many early studies on Ub-binding receptors were in agreement with this hypothesis. For example, initial studies on the UBA domains from the UbL-UBA shuttles proposed that Dsk2/Dph1 from yeasts (in *S. cerevisiae* and *S. pombe*, respectively) and DSK2A-B from the *Arabidopsis* preferentially bind to Lys48-linked Ub chains, rather than to Lys63-linked Ub chains or mono-Ub [85,87,177]. Human HR23A also bind Lys48-linked Ub chains [83]. Similarly, the plant RAD23 family also has a strong preference for proteins marked by Lys48-linked Ub chains [178]. This led to the notion that the primary, Ub-based, proteasomal degron is the Lys48-linkage and UbL-UBA proteins such as yeast Dsk2, Rad23 and Ddi1, as well as their homologues in higher eukaryotes are proteasomal shuttling receptors.

Conversely, early studies on UBA domains of SQSTM1 and NBR1 showed that these proteins preferentially bind Lys63-linked Ub chains. The NBR1 UBA domain binds to Lys63-linked di-Ub about 60 times stronger than to mono-Ub [30]. Similar results have been found for the UBA domain of SQSTM1, which prefers binding to Lys63-linked Ub chains, but still has a lower affinity for Lys48-linked Ub chains [139,179,180]. On the basis of these experiments, it was proposed that Lys63-linked Ub chains mark proteins for autophagic clearance.

However, increasing evidence is accumulating to suggest that the Ub recognition code is inadequate to explain how a decision is made between proteasomal degradation and autophagic turnover as alternative PQC pathways. For example, further studies on the UBA domain of NBR1, which eliminated an artifact baused by dimerization of the glutathione-S-transferase tag fused to it, have shown that it lacks specificity for poly-Ub linkages [147]. Specifically, the UBA domain of NBR1 has no preference for Lys63-linked or Lys48-linked di-Ub and binds to each monomeric unit of Lys48-linked and Lys63-linked poly-Ub with similar affinities and via the same surface as its binding to mono-Ub [147]. It is especially intriguing in the context of this recent finding that Ub-positive inclusions do not show any linkage specificity, which demonstrates that Lys63-linked poly-Ub is not an exclusive and definitive label for autophagy [181].

New studies on yeast Dsk2 also require modifications of the previous views on the prevalent signals for proteasomal degradation. The Dsk2 UBA domain has high affinity for both mono-Ub and poly-Ub chains, but no substantial preferences for Lys48-linked or Lys63-linked Ub chains [90]. Similarly, the UBA domain of ubiquilin1 shows little or no binding selectivity toward a particular chain linkage, or between the two Ub moieties in the same chain [101].

Moreover, recently multiple Lys48-linked di-Ub conjugates have been proven to serve as stronger degrons for proteasomal degradation than tetrameric Lys48-linked poly-Ub chains, which were believed to be the most efficient proteasomal degron [182,183]. Additionally, recent studies indicate that other linkages could also be recognized by the proteasomal and autophagic machineries [34,175,184]. For instance, Lys11-linked Ub chains trigger proteasomal degradation of cell cycle regulators during mitosis [185,186]. In addition, but less frequently, Lys29-linked and Lys63-linked Ub chains mediate proteasomal degradation and reveal the plasticity in the recognition of Ub labels by degradation machineries [187,188,189,190,191]. These variations mean that the Ub code on certain substrates may not uniquely determine which degradation path they will follow. Almost any type of Ub chain: homogenous, heterogenous, linear, single, and multibranched chain, as well as mono-Ub and multi-mono-Ub label can be accepted by proteasomes [34]. For instance, SQSTM1, which delivers primarily ubiquitinated cargoes for autophagic degradation, also shuttles Lys63-linked poly-ubiquitinated substrates for proteasomal degradation. The UBA domain of SQSTM1 recognizes Lys63-linked poly-Ub chains of tau and mediates the interaction between Lys63-linked, poly-ubiquitinated tau [192] and the PSMC2 (proteasome 26S subunit, ATPase 2) homologous to yeast Rpt1 subunit of proteasomes, which is a docking site for shuttling factors. Similarly, SQSTM1 is involved in the proteasomal turnover of Lys63-linked, poly-ubiquitinated TrkA, also known as NTRK1 (neurotrophic receptor tyrosine kinase 1) [132]. In this regard, the function of SQSTM1 is similar to that of other proteins containing UBA and UbL domains, such as yeast Ddi1, Rad23, and Dsk2 (and their homologues in higher eukaryotes) in shuttling of substrates to the proteasome. However, instead of delivering proteins marked by Lys48-linked poly-Ub chains, substrates with Lys63-linked Ub chains are targeted to the proteasomes.

Recent studies on human and plant homologues of Dsk2 reiterate that shuttling of SQSTM1 between both degradation systems is not an exception among Ub-binding proteins. While human HR23 is still a major receptor for the proteasome pathway, ubiquilins function in several distinct protein degradation pathways by delivering substrates to either the proteasome, autophagy, or ERAD pathways [97]. In response to accumulation of aggregation-prone proteins, ubiquilin1 with its cargo is recruited by epidermal growth factor substrate 15 (EPS15) to the MTOC, where aggresomes form [94]. These results have been explained in terms of a model that proposes a dual role for ubiquilin1, whose binding to the UIM domains of the proteasomal subunit S5a and the EPS15 protein is mutually exclusive [94]. Thus, when there are low levels of protein aggregation, ubiquilin1 would promote the shuttling of poly-ubiquitinated proteins to the proteasome. However, when the level of protein aggregation is high and proteasomes may be running at capacity, ubiquilin1, via its UbL domain, can alternatively interact with EPS15. Therefore, this promotes aggresome formation. However, it is unknown if ubiquilin1-EPS15 interaction is evolutionarily conserved. Consistently, it has been reported that yeast Dsk2 facilitates the vacuole-mediated clearance of misfolded proteins by promoting inclusion body formation [193] and loss of the yeast homologue of EPS15, Pan1, enhances protein aggregate toxicity [194].

Ubiquilin2 and ubiquilin4 also have a role in autophagy. Their knockdown renders cells more susceptible to starvation-induced cell death while overexpression has a protective effect [96]. Ubiquilin2 co-localizes and co-immunoprecipitates with LC3 and assists in the maturation of autophagosomes to autolysosomes [95,96,195]. Additionally, ubiquilin4 directly associates with LC3 and, as an LC3-interacting partner, links the cargo-carrying ubiquilin1 to the autophagy machinery [196].

Other work has clearly indicated that plant DSK2, which associates with Lys48-linked Ub chains more strongly than with Lys63-linked Ub chains [177], when properly phosphorylated, serves as an autophagic receptor for bri1-EMS-suppressor 1 (BES1) protein due to the enhancement of its interaction with the autophagy protein ATG8 [197].

Lastly, recent results show that ubiquilin2 acts with the HSP70-HSP110 (heat shock 70 kDa protein—Heat shock 110 kDa protein) disaggregase machinery to deliver aggregated and misfolded proteins to the proteasome for their degradation [198] and, therefore, mediates autophagy-independent protein aggregate clearance by the proteasome and adds another layer of complexity in the fate of ubiquitinated proteins [198]. In the model proposed by these authors, under non-stressed conditions, ubiquilin2 is held inactive as homo-dimers or hetero-dimers, but the presence of HSP70 and ubiquitinated misfolded/aggregated proteins activates ubiquilin2, which then associates with HSP70 [198]. Next, HSP70/HSP110-dependent disaggregase activity pulls aggregated proteins apart, which allows ubiquilin2 to drive ubiquitinated misfolded proteins to the proteasomes for degradation.

Together, these results expose new complexities in the regulated proteolytic turnover of ubiquitinated substrates. Autophagic receptors shuttle between proteasome and autophagy pathways and proteasomal receptors might be tied to both degradation mechanisms. Since bi-directionality (towards proteasomes and autophagy) of Ub-binding shuttles appears to be a common mechanism that links the autophagy pathway and the UPS, previous models suggest that the Ub code alone determining this fate is not sufficient. It must be noted here that the Ub code indubitably plays an important role in the initial pre-commitment process. However, in light of recent data, the linkage of the accurate Ub label is insufficient to assign a protein substrate irreversibly to particular degradation pathway. These facts raise the question regarding which other factors may affect the PQC pathway choice.

## 6. PQC Pathway Choice Based on an Oligomeric State of the Receptor and Recognition of ATG8/LC3/GABARAP Proteins

To unravel the crucial determinants for the PQC pathway choice, a set of modular artificial receptors harboring identical Ub-binding modules, but with various oligomerization potentials, was created [90]. These modules either included or excluded an AIM/LIR and/or the UbL domain. Monomeric and dimeric forms of artificial receptors that contained the UbL domain (regardless of whether they possessed an AIM or not) supported efficient proteasomal degradation of soluble proteins but failed in clearing aggregated proteins. Conversely, oligomeric receptors containing an AIM (but not the UbL) supported autophagy-dependent degradation of aggregates but were impaired in the proteasomal clearance of soluble proteins. Thus, it has been clearly demonstrated that the PQC pathway choice can still be made at the level of the receptor after the initial substrate ubiquitination by competition between Ub-binding receptors harboring either proteasome-binding or ATG8-binding modules [90]. Therefore, a predisposition to organize higher oligomers and to bind ATG8 seems to be essential for autophagic degradation.

All known autophagic receptors involved in aggregate clearance, such as SQSTM1, NBR1, OPTN and others described above, can oligomerize and interact (directly or indirectly) with ATG8/LC3/GABARAP family members. For instance, SQSTM1 polymerizes via its PB1 domain into helical filaments and even assembles into higher order hetero-oligomers with different proteins like autophagy-linked FYVE protein (ALFY), a BEACH domain– and WD40-containing protein (WDR81), or Huntingtin (HTT), which additionally stimulate SQSTM1-mediated clustering of aggregated proteins and their autophagic turnover [26,137,199,200,201]. NBR1 self-oligomers via its CC1 domain and forms hetero-oligomers with SQSTM1 by PB1-PB1 interactions to render SQSTM1 clustering more efficient [27]. OPTN also oligomerizes and interacts with SQSTM1 [202].

Oligomerization of SQSTM1 is a factor that generates high-avidity interactions with Ub and the LC3/GABARAP proteins and, therefore, initiates autophagic turnover of substrates bound to SQSTM1. Oligomerization of SQSTM1 promotes the interaction with Ub and LC3B by drastically increasing the residence time of SQSTM1 on LC3B and Ub-coated aggregates [139], which may, perhaps, allow aggregates to be delivered by SQSTM1, together with the Ub-binding receptor, to autophagosomes and, subsequently, to lysosomes [139]. Consistently, deletion of the PB1 domain of SQSTM1, or mutations that interfere with its ability to multimerize, inhibit the recruitment of SQSTM1 to autophagosomes [138].

With regard to receptor binding to ATG8/LC3/GABARAP proteins, in some Ub-binding receptors, the AIM/LIR motif may be cryptic, and require activation by phosphorylation. Binding to ATG8/LC3/GABARAP proteins family members might also be indirect via other proteins that mediate receptor binding to the autophagosome membrane. For example, the function of ubiquilin1 as an autophagic carrier is mediated through ubiquilin4, which recognizes LC3 via two STI1 repeats lacking a canonical LIR [196].

## 7. PQC Pathway Choice Based on Other Factors

### 7.1. Chaperone Assistance in the Pathway Choice

The best example of the role of chaperones in modulating and regulating the PQC pathway choice comes from the studies on the Bcl2-associated athanogene (BAG) family of proteins in mammals, which are implicated in the coordination of proteasomal and autophagic activation. BAG1 is a well-known HSP70 chaperone that recognizes poly-ubiquitinated misfolded or unfolded substrates and initiates their sorting for proteasomal degradation [203,204]. However, in case of proteasome impairment or overload caused by stress or aging, another member of BAG family, BAG3, promotes turnover of poly-ubiquitinated proteins by autophagy [205,206,207,208]. BAG3 not only induces expression of LC3B and SQSTM1, but also directly interacts with SQSTM1, redirecting for autophagic degradation proteins labeled by Lys48-linked Ub chains that may originally have been destined for proteasomal clearance [209]. Moreover, BAG3, by interacting with the dynein motor and 14-3-3 proteins, actually induces the sequestration of poly-ubiquitinated proteins into aggresomes at the MTOC [210]. In summary, when cells switch their expression form BAG1 to BAG3, they also switch PQC systems from BAG1-mediated proteasomal degradation to BAG3-mediated autophagy.

Since all studies on the BAG1/BAG3 ratio and BAG3-mediated autophagy have come from the animal kingdom [206], it is worth asking if such modulation of the PQC system is also present in plants where the BAG protein family is conserved [211,212,213,214,215].

### 7.2. Conformational Changes of the Ub-Binding Receptors—The Intramolecular and Intermolecular Interactions

A number of proteins that contain both UbL and UBA domains, including the Ub-binding receptors involved in proteasomal degradation, form intramolecular interactions between the UbL and UBA domains. These domains from Rad23 and Dsk2 display intramolecular interactions even though the binding affinity was low [216]. Similar results have been obtained for human HR23A [75] and HR23B [76]. Such an interaction is possible due to adoption of an Ub-like fold by the UbL domain, which is recognized by the UBA domain. Studies on HR23A have revealed that UbL domain binding to the UBA domain and to proteasomal subunit S5a are mutually exclusive and HR23A loses its interdomain structure as a consequence of S5a binding [75]. At the same time, binding of the HR23A UBA domain to Ub also structurally alters the rest of the HR23A protein, wherein residues within the UbL domain that reside in the UBA contact surface shift. This suggests that binding of Ub to the UBA domain would prevent the intramolecular interaction with the UbL domain [217]. Based on these results, it seems reasonable that in the absence of proteasomal subunits, or Ub substrates that could interact with the UbL and UBA domains, respectively, intramolecular interaction between these domains can “lock” these proteins into a “closed” conformation. Because Dsk2 and Rad23 dimerize via their UBA domains [104,218], “closed” conformations would affect their homo-dimerization. However, not all UbL-UBA shuttles dimerize through their UBA domain. Ubiquilin1 mutants deleted for either their UbL or UBA domain retain the ability to interact with each other, which indicates that the central region is required for the interaction [219]. Similarly, Ddi1 can form homo-dimers even after the deletion of the UbL or UBA domains, which indicates that their dimerization is also mediated by the central region [104]. Therefore, it is an open question if adoption of such “closed” conformation would affect dimerization of these proteins.

Additionally, intermolecular interactions between these domains have also been detected. For instance, the UBA domain of one of the human Dsk2 homologues, ubiquilin2, can interact with the UbL domain of HR23Aas well as UBA2 domain of HR23A can interact with the UbL domain of ubiquilin2 [220]. Because intermolecular UbL-UBA binding should prevent the intramolecular UbL–UBA interaction in HR23A, it is possible that competition between intra-molecular and intermolecular UbL-UBA interactions occurs, which leads to the formation of shuttle monomers or multimers.

The NMR studies of the PB1 domain have shown that it creates an Ub-like, β-grasp fold, similar to the UbL domain [221]. Therefore, it has been postulated that the PB1 domain also directly interacts with its UBA [222,223]. Weak interaction between PB1 and UBA domains has been detected for the plant homologue of SQSTM1, Joka2 [148]. The Joka2 PB1-UBA interactions were only observed in aggregates, which suggests that clustering of Joka2 into higher order homo-oligomers requires PB1 oligomerization, as well as PB1-UBA interactions [148].

Thus, it appears that all proteins with UbL/PB1 and UBA domains possess the ability to dimerize/oligomerize through either the PB1 or UBA domains or the central region of the protein, and can form other intramolecular UbL/PB1-UBA interactions that could affect their predisposition to aggregation.

### 7.3. Post-Translational Modifications of the Ub-Binding Receptors

Post-translational modification is a common mechanism to modify protein properties. Additionally, when such modifications affect protein localization or function, it adds further flexibility to protein utility. Since Ub-binding protein shuttles cooperate with both proteasomal and autophagic PQC machineries, it can be expected that their involvement in these pathways should be regulated via various post-translational modification(s). Many shuttling receptors undergo specific modifications to properly respond to the cellular needs (Table 1). In this section, we discuss those modifications that might influence the decision-making process between autophagy and the UPS. For example, it has recently been reported that SQSTM1 is ubiquitinated at Lys7 within its PB1 domain by TRIM21, which negatively affects its oligomerization and sequestration activity [224]. Unfortunately, it is unknown if this has any effect on its involvement in proteasomal clearance (since PB1 binds Rpt1 and Rnp10 subunits of proteasome).

Interestingly, SQSTM1 can be ubiquitinated within its UBA domain at the highly conserved Lys420 by KEAP1/CUL3 (Cullin-3). Modifications mediated by KEAP1/CUL3 increase SQSTM1 sequestering activity, association with LC3B, and its final degradation by affecting UBA dimerization [244]. It is unclear if this modification abrogates self-association of the UBA domain to increase Ub binding (simultaneous dimerization and Ub binding are mutually exclusive) or whether interaction of the UBA domain with other proteins promotes SQSTM1 oligomerization. However, it is clear that different ubiquitination statuses of SQSTM1 can positively or negatively affect its role as the autophagic receptor and, thereby, alter the fate of ubiquitinated substrates (for instance, TRIM21 ubiquitination would antagonize KEAP1/CUL3 mediated ubiquitination at the C-terminal UBA domain).

SQSTM1 is subject to phosphoregulation as well. Phosphorylation of Ser403 and Ser409 within the UBA domain of SQSTM1 destabilizes self-association of this domain and, therefore, positively regulates SQSTM1 activity [229,237]. It is suggested that phosphorylation at Ser403 increases SQSTM1 affinity for Ub by enabling the formation of additional polar contacts with Ub, analogous to polar contacts of the UBA domains of NBR1 and Dsk2, formed by Glu926 and Asp341, respectively, with the side chains of Lys6 and His68 in Ub. Because NBR1 and Dsk2 permanently possess negative residues upstream of the conserved MGF motif, which plays an integral role in Ub recognition, their binding to Ub is always stronger than that of the unphosphorylated UBA in SQSTM1 [147]. Intriguingly, around 17% of all known UBA domains have Ser residue in a position corresponding to Ser403 in SQSTM1, which suggests that phosphorylation might be a wide-ranging mechanism of attenuation of Ub-binding affinity in UBA domains and the subsequent activation of these proteins to recognize Ub conjugates.

Phosphoregulation of the AIM/LIR within Ub-binding receptors is another decisive post-translational modification that impacts PQC pathway choice by shuttling receptors. The canonical AIM/LIRs are represented by the W/F/F-X-X-L/I/V consensus, which is often adjacent to acidic or phosphorylated residues that form additional contact sites with members of the ATG8/LC3/GABARAP protein family [245,246]. Human SQSTM1 and NBR1, as well as its NBR1-like counterpart from *A. thaliana*, already contain negatively charged residue(s) just upstream of the core AIM motif, making them receptors already “primed” for autophagy [24]. However, about 25% of known AIM/LIR motifs harbor Ser or Thr residues at that position suggest that functionality of these AIM/LIRs might be regulated through phosphorylation. For instance, plant DSK2, when phosphorylated by the glycogen synthase 3 (GSK3)-like kinase, BIN2 (brassinosteroid insensitive 2), near its AIMs, targets BES1 for autophagic clearance. In other words, the targeting of BES1 for autophagic degradation is possible due to the DSK2-ATG8 interaction mediated by the activation of cryptic AIMs by BIN2 phosphorylation [197].

In contrast, phosphorylation of NBR1 by GSK3 at Thr586 within its non-canonical LIR inhibits its adaptor function in the formation of proteins aggregates and, therefore, prevents the aggregation of poly-ubiquitinated proteins and their subsequent selective degradation [242].

In summary, posttranslational modifications can modulate the functional status of Ub-binding shuttles and, therefore, can affect the pathway choice between the UPS and autophagy.

## 8. Suggested Model and Concluding Remarks

Proteasomal degradation and autophagy are two independent elements of the PQC system, but they act in a networked and interactive manner to maintain proteostasis. Multiple lines of evidence presented above suggest that Ub-binding shuttles can play roles as integration centers of both systems, which, thereby, coordinates the communication between the UPS and autophagy in the clearance of protein aggregates. The old model, focusing primarily on the Ub code, is incapable of explaining clearly the pathway chosen by a given ubiquitinated substrate. In view of new available data, we propose a model in which the Ub-binding receptors along with their associated regulators (interacting partners, chaperones, proteins conferring post-translational modifications) and the Ub code integrate the complex hierarchy of decisions regarding how unwanted proteins will be degraded.

In this model, both systems known as the UPS and autophagy share the ubiquitination process that label substrates. This is the first step where a pre-commitment decision can be made regarding which system will be used for protein clearance (Figure 1 and Figure 3). There is evidence demonstrating that labeling proteins with particular Ub codes already assigns this protein to a particular degradation system. Thus, the original model built on the conviction that the Ub code stores information about protein fate may still be correct, but refers mostly to optimal conditions and fully operating proteasomes. Because cells must handle constantly changing environments and respond to changes instantly, further “safe alternatives” from pre-commitment decisions are required. While the E2/E3 ligase complexes “write” the Ub code and attach them to the substrate, other deubiquitinating enzymes (DUBs) “erase” the code by removing Ub linkages (Figure 3). This mechanism of editing of previously written Ub codes enables the adjustment of Ub signaling to cellular needs.

In principle, within the Ub code model, it is plausible that a code on a given protein could be erased and rewritten to shift the balance from one degradation pathway to another. However, the current evidence does not favor the rewriting of the Ub code as a major mechanism to redirect proteins to the alternative degradation pathway. Thus, after ubiquitination per se, additional decision-making steps must play a role in determining, for example, which Ub-binding adaptor proteins (such as receptors with PB1/UbL and UBA domains), will be used to recognize the Ub code on the target substrate, and which of the two PQC machineries these adaptors will engage with (Figure 3).

As we try to show in this manuscript, the role of these proteins in proteasomal and autophagic clearance is complex and has been an area of active debate and is under continued revision. A new appreciation is the role of protein modifications in regulating the function of Ub-binding shuttles and the effect of these on the PQC pathway choice. Numerous modifications within the UBA domain affect the affinities of these receptors for Ub or modulate the self-association of these receptors. Similarly, many post-translational modifications of the PB1 or UbL domain receptors impact their dimerization or interaction with proteasomal subunits. We also assume that both intra-molecular and inter-molecular interactions of PB1/UbL and UBA domains are part of the mechanism shared by these receptors to regulate their function by dynamically changing conformation from “closed“ to “open” or from monomeric to multimeric states and vice versa (Figure 4). Lastly, the interactions of these receptors with the autophagy machinery can also be regulated by phosphorylation of AIM/LIRs.

It is plausible that, under un-induced conditions, the receptors with PB1/UbL and UBA domains could reside in a “closed” conformation where the UBA domain binds to the PB1/UbL domain. In such a state, the receptor may be shielded from binding ubiquitinated substrates as well as the proteasome or from oligomerizing and clustering into larger structures. However, because PB1/UbL binding to UBA is rather weak, more favorable conditions might activate the receptors and promote an “open” conformation. In such a case, the PB1 or UbL domains could bind the proteasome or other binding partners (either other proteins or another PB1 domain leading to oligomerization), which allows the UBA domain to homo-dimerize or to interact with ubiquitinated substrates, according to the ubiquitination/phosphorylation status of the UBA. While the molecular basis of the receptor conformation shifts remains unclear, the concentration of free Ub, ubiquitinated proteins, or the activity of particular E3 ligases and DUBs as well as coordinated phosphorylation/dephosphorylation are the most reasonable candidates for receptor conformation regulation (Figure 4).

Many reports show that ubiquitination or phosphorylation of certain residues affect receptor recognition by proteasome or its tendency to multimerization (Table 1). High concentrations of free mono-Ub and unanchored Ub chains inhibit clustering of SQSTM1, which might be related to the Ub-binding activity of the ZZ domain and might sterically inhibit oligomerization of SQSTM1 [27]. However, in contrast to Ub binding, the interaction of the N-arginylated chaperone, BiP (binding immunoglobulin protein), with the ZZ domain induces a conformational change of SQSTM1 exposing its PB1 and LIR, which accelerates its self-oligomerization and targeting to autophagosomes [247]. Since N-terminal arginylation of BiP is induced by accumulation of ubiquitinated proteins and aggregates, this serves as an interesting example of how cells control the crosstalk between the UPS and autophagy in decision-making process and react to accumulating proteins.

In conclusion, the clearance of unwanted and misfolded proteins is crucial for the maintenance of cellular proteostasis. Although the PQC network targeting ubiquitinated abnormal or unwanted proteins for degradation is hierarchically organized, it must retain flexibility and coordinate the two PQC pathways, according to the needs. Under normal conditions, unwanted proteins initially become substrates for the UPS and are primarily recognized by proteasomal receptors (such as Rad23 or Dsk2) characterized by higher affinity to Ub than that by autophagic receptors. Then, simultaneous binding of such receptors to proteasomal proteins via the UbL domain on the receptors and the substrates permits the transfer of the degradation substrate to the proteasome (Figure 3). However, when proteasomes are overloaded or malfunctioning, which could also be affected by stress, ubiquitinated proteins accumulate and form aggregates on which Ub is locally concentrated. Such structures can be recognized by autophagic receptors, such as SQSTM1, which, as a monomer, has low affinity, but as an oligomer, has high avidity to Ub and spontaneously coalesces with ubiquitinated proteins into larger clusters (Figure 3). Aggregation or aggresome formation can additionally be promoted by interaction with other oligomerizing and scaffolding proteins such as NBR1 [29,142], OPTN [32,115,116], or ALFY [199,200,248,249], which increase SQSTM1-mediated clustering [27,29,30,137]. In addition, the LIR motif sequence by itself, in contrast to its ability to bind to LC3B, promotes aggregation and delivery to the autophagic machinery [27]. However, in times of cellular need, shuttling receptors mainly assigned to the UPS or autophagy can adapt to the alternative such as a functional, degradation system in support of cellular proteostasis (Figure 3). We further hypothesize that the impact of shuttling Ub-binding receptors on the fate of ubiquitinated targets relies on the combined effect of several regulatory factors acting on those receptors: affinity to particular type(s) of Ub chains, intramolecular and intermolecular homo-oligomeric and hetero-oligomeric interactions of each domain present in these receptors, exposition of LIR/AIM, and post-translational modifications affecting protein conformational changes (Figure 4).

## 9. Future Perspectives

It has become clear that the UPS and autophagy are major PQC systems for the degradation of unwanted proteins in the cytoplasm. It has now been generally acknowledged that these two systems are highly interlinked and accept similar Ub labels on their targets. Moreover, based on recent studies, it appears that there is no single, specific signal that targets substrates exclusively to either the UPS or autophagy. The great majority of ubiquitin–UBA analyses have been done as in vitro studies and are often contradictory to the analyses of Ub linkages found in vivo, which include contributions of other cellular factors in Ub-linkage maintenance and selectivity. Therefore, it seems that revision of previous data and new careful analyses need to be done for receptor with UbL/PB1 and UBA domains to determine their predisposition to particular types of Ub conjugates in particular conditions. The physiological relevance of Ub linkages other than Lys48-linkages and Lys63-linkages or more complex structures, such as branched chains, remains open for additional research. We still do not know how dynamic and flexible Ub chains are and if their structure can be regulated to modulate recognition of particular Ub-binding receptors. Moreover, we are just beginning to understand how Ub-binding shuttles are regulated and how this regulation affects which degradation pathway they choose. Furthermore, an additional layer of complexity in the fate of ubiquitinated proteins comes from the fact that UbL fold can be found in numerous proteins that serve other unrelated functions within the cell. These important questions promise exciting times ahead for this field of research.

## Figures and Tables

**Figure 1 cells-08-00040-f001:**
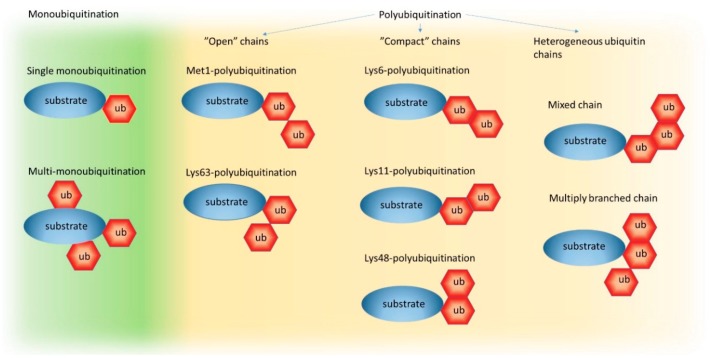
The ubiquitin code. A schematic representation of possible ubiquitin (Ub) chain formations on a target protein (substrate) from single mono-ubiquitination to mixed and multiply -branched Ub chains. Within polyubiquitin chains, which can exist on one of the seven intrinsic lysine (Lys) residues within the Ub sequence, or methionine (Met) at position 1, Ub can form eight different homogeneous or heterogeneous linkage types. However, only five linkage types have been structurally characterized (Met1-chains, Lys6-chains, Lys11-chains, Lys48-chains, and Lys63-chains) and classified as “open” or “compact” based on their three-dimensional topologies.

**Figure 2 cells-08-00040-f002:**
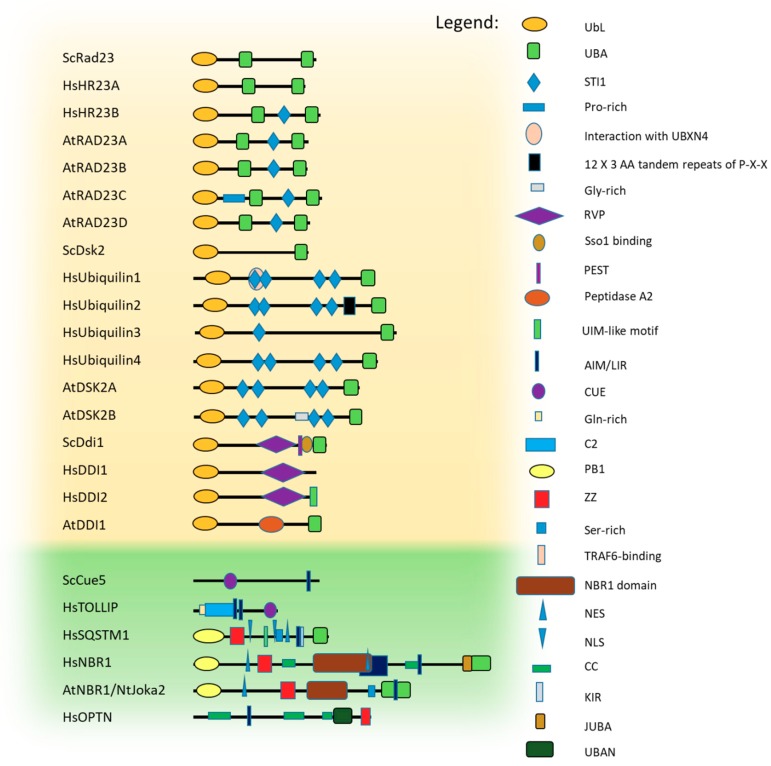
Ub-binding protein shuttles from yeast, mammals, and plants. Schematic representation of the domain/motif/region organization of the known Ub-binding shuttle receptors used by the proteasomal [59] and autophagic [24,65,66,67] protein quality control (PQC) pathways. The graphical depiction of Ub-binding modules is based on their identification using UniProt and extended by data in the published literature [68,69]. The domain/motif/regions are shown for the proteasomal (*S. cerevisiae* Rad23 (P32628), *H. sapiens* HR23A (P54725), *H. sapiens* HR23B (P54727), *A. thaliana* RAD23A (Q84L32), *A. thaliana* RAD23B (Q84L33), *A. thaliana* RAD23C (Q84L31), *A. thaliana* RAD23D (Q84L30), *S. cerevisiae* Dsk2 (P48510), *H. sapiens* ubiquilin1 (Q9UMX0), *H. sapiens* ubiquilin2 (Q9UHD9), *H. sapiens* ubiquilin3 (Q9H347), *H. sapiens* ubiquilin4 (Q9NRR5), *A. thaliana* DSK2A (Q9SII9), *A. thaliana* DSK2B (Q9SII8), *S. cerevisiae* Ddi1 (P40087), *H. sapiens* DDI1 (Q8WTU0), *H. sapiens* DDI2 (Q5TDH0), *A. thaliana* DDI1 (Q1EBV4)), and autophagic shuttles (*S. cerevisiae* Cue5 (Q08412), *H. sapiens* TOLLIP (Q9H0E2), *H. sapiens* SQSTM1 (Q13501), *H. sapiens* NBR1 (Q14596), *A. thaliana* NBR1 (Q9SB64), *N. tabacum* Joka2 (F8RP79), and *H. sapiens* OPTN (Q96CV9)). All proteins and domains are drawn to scale. UbL, Ubiquitin-like. UBA, Ubiquitin-associated. STI1, stress-inducible 1. UBXN4, Ubiquitin-regulatory X domain-containing protein 4. RVP, retroviral protease fold domain. PEST, protein rich in Pro, Glu, Ser and Thr. UIM, Ub-interacting motif. AIM/LIR, ATG8-interacting motif/LC3-interacting region. CUE, coupling of Ub-conjugation to ER degradation. C2, Ca^+2^-dependent membrane-targeting module. PB1, Phox and Bem1. ZZ, Zinc finger. TRAF6, TNF receptor-associated factor 6. NBR1, Neighbor of BRCA1. NES, Nuclear Export Signal. NLS, Nuclear Localization Signal. CC, coiled-coil. KIR, KEAP1 (Kelch-like ECH-associated protein 1)-interacting region. JUBA, Juxta-UBA. UBAN, UBD Ub-binding in A20-binding inhibitor of NF-κB (ABIN) proteins and NF-κB essential modulator (NEMO).

**Figure 3 cells-08-00040-f003:**
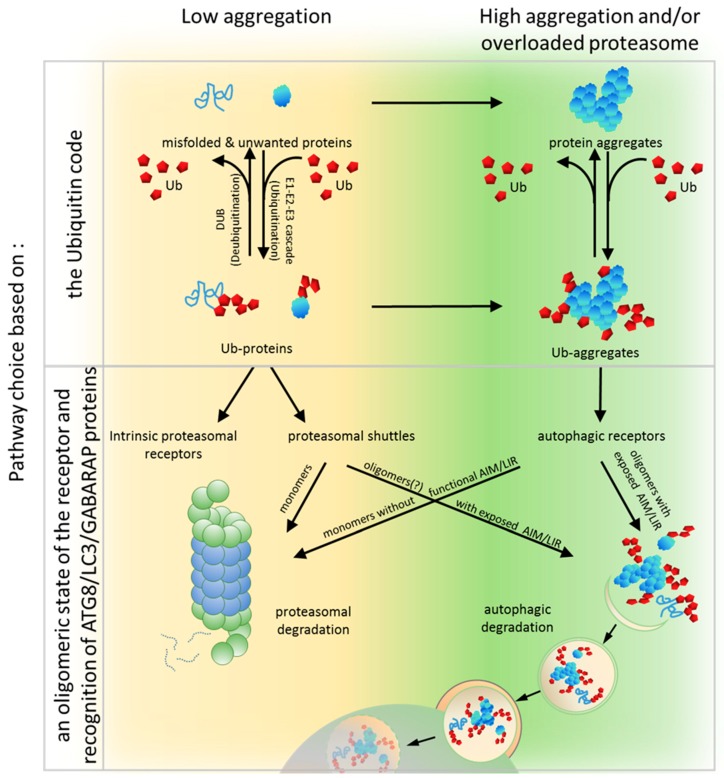
A model for the substrate fate based on the Ub code and an oligomeric state of the receptor and recognition of ATG8/LC3/GABARAP (autophagy-related protein 8/microtubule-associated protein 1A/1B-light chain 3/GABA type A receptor-associated protein) proteins. The PQC network targeting misfolded or unwanted proteins via ubiquitination for degradation is hierarchically organized, but retains flexibility and coordinates the Ub-proteasome system (UPS) and autophagy pathways, according to cellular needs. In the low aggregation (undisturbed) condition, unwanted proteins initially become substrates for the UPS. During ubiquitination, particular Ub labels are attached to the target proteins via the E1–E2–E3 enzyme cascade. This process is reversed by deubiquitinases (DUBs). When the Ub code is written on the substrates, the cargos are then primarily recognized by intrinsic or shuttling proteasomal receptors characterized by higher affinity to Ub in comparison to the affinity of autophagic receptors to Ub. Simultaneous binding of shuttle receptors (probably in monomeric states) to ubiquitinated cargo and proteasomal proteins permit the transfer of the degradation substrates to the proteasome. However, in high aggregation conditions such as when proteasomes are overloaded or when misfolded or redundant proteins accumulate and form aggregates on which Ub is locally concentrated. Such aggregates are mainly recognized by autophagic receptors, which in their oligomeric states have high avidity for Ub and spontaneously coalesce with ubiquitinated proteins into larger clusters that further promote substrate aggregation and delivery to the autophagic machinery. In times of cellular need, shuttling receptors from the proteasomal or autophagic clearance pathways adapt to the alternative, but functional, degradation system to support cellular proteostasis. Proteasomal shuttles with exposed AIM/LIR motif and/or with a tendency to oligomerization can support protein turnover via autophagy by directly driving substrates to autophagic machinery or indirectly by sequestering ubiquitinated proteins in aggregates and stimulating cluster formation. Conversely, autophagic receptors in monomeric forms with non-functional AIM/LIR motifs can promote proteasomal degradation, if they are able to recognize and engage proteasomal subunits.

**Figure 4 cells-08-00040-f004:**
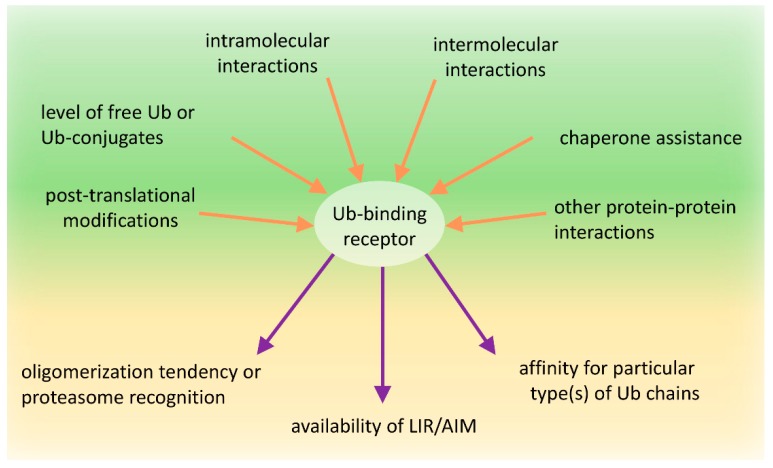
Schematic representation of factors affecting conformational changes and function of Ub-binding receptors. The impact of shuttling Ub-binding receptors on the fate of ubiquitinated targets relies on the combined effect of several regulatory factors acting on those receptors and affecting their oligomerization tendency or proteasome recognition, availability of AIM/LIR motifs for ATG8/LC3/GABARAP-family member proteins decorating isolation membranes, and the receptor affinity for particular type(s) of Ub chains. Factors influencing these choices include post-translational modifications, intramolecular, intermolecular homo-oligomeric, and hetero-oligomeric interactions of each domain present in these receptors, chaperone assistance, levels of available free Ub and Ub-conjugates and effects of other proteins and molecules interacting with Ub-binding receptors and affecting their conformational changes and/or function.

**Table 1 cells-08-00040-t001:** Ub-binding receptors involved in protein quality control (PQC) pathways.

Species	Ub-Binding Protein	Modification/s	Function	Reference/s
Yeast	Rad23	S47 and S73 phosphorylation	Inhibits binding of Rad23 to proteasome	[225]
UbL deubiquitination by Ubp12	Stabilizes Rad23-substrate binding by inhibiting proteasomal degradation of substrate	[226]
Dsk2	ubiquitination	Reduces capacity to bind poly-ubiquitinated proteins	
Ddi1	No data		
Cue5	No data		
Human	HR23B	No data		
Ubiquilin1-4	No data		
Ddi1	No data		
TOLLIP	Phosphorylation by IRAK1	Regulates TLR-mediated cell activation by dissociation of IRAK1 from TOLLIP	[227]
SQSTM1	K7 ubiquitination by TRIM21	Abrogates oligomerization via PB1 domain and inhibits sequestration activity	[224]
K13 ubiquitination by Parkin	Promotes the proteasomal degradation of SQSTM1	[228]
S24, S207, S403 phosphorylation	Role unknown, probably autophagy stimulation as a response to the MG132 treatment	[229]
K91 and K189 ubiquitination by RNF166	Mediates xenophagy, K29- polyubiquitination and K33-polyubiquitination of SQSTM1	[230]
T138 phosphorylation by LRRK	Role unknown, increases the neurotoxicity	[231]
T269 and S272 phosphorylation by CDK1	Regulates exit from mitosis	[232]
S294 phosphorylation by AMPK	Induces mitophagy and autophagic cell death	[233]
S349 phosphorylation by CK1, TAK1	Enhances the binding affinity between KEAP1 and SQSTM1, activates the KEAP1-NRF2 pathway	[234,235,236]
S403 phosphorylation by ULK1, CK2, TBK1	Attenuates affinity to Ub	[229,237,238]
S407 phosphorylation by ULK1	Attenuates affinity to Ub, facilitates phosphorylation of S405	[233,234,237]
K420 ubiquitination by KEAP1/CUL3	Increases sequestering activity and degradation	[233,234,237,239]
UBA domain ubiquitination by RNF26	Facilitates vesicular cargo sorting	[239,240]
PB1 domain ubiquitination by NEDD4	Facilitates inclusion body formation and autophagy mediated by SQSTM1	[241]
NBR1	T586 phosphorylation by GSK3	Inhibits aggregation of ubiquitinated proteins and their degradation	[242]
OPTN	S177 phosphorylation by TBK1	Activates LIR motif and promotes LC3 binding	[243]
S473 and S513 phosphorylation by TBK1	Promotes poly-Ub binding	[243]
S473 phosphorylation	Promotes poly-Ub binding during mitophagy	[114]
S177 and S513 phosphorylation by IKKβ	Role unknown probably promoting autophagy	[158]
Plant	RAD23A-D	No data		
DSK2A-B	Phosphorylation of sequences adjacent to AIMs by BIN2	Promotes the autophagic degradation of BES1	[197]
DDI1	No data		
NBR1	No data

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
