# Peer review of "The Roles of Ubiquitin-Binding Protein Shuttles in the Degradative Fate of Ubiquitinated Proteins in the Ubiquitin-Proteasome System and Autophagy"

_cells, 2019, doi:10.3390/cells8010040_

Round 1
Reviewer 1 Report
Review report about:
Title
The roles of ubiquitin-binding protein shuttles in the degradative fate of ubiquitinated proteins in the ubiquitin-proteasome system and autophagy
Authors
Katarzyna Zientara-Rytter , Suresh Subramani
The authors did a great job collecting and evaluating the current literature about ubiquitin binding proteins and ubiquitin mediated proteolysis. The reader is well introduced into the topic and the chapters are in a logic order. The authors stick to known facts and do not spend space for lengthy speculations. Bothe expert readers, and scientist new to this exciting field, can extract many information from this nice data compilation. All relevant literature is cited and critically discussed if necessary. The authors avoided to deflect the attention of the reader from the main message, because unnecessary details are spared. The figures (despite the background color) and tables are easy to understand without reading the whole text and represent a successful aggregation of the text. The authors also give particular attention to plant proteins, which are ignored by many other scientists. The whole text is in excellent English.
Author Response
To improve the clarity of the text and to keep the reader's focus on the main scope of this manuscript, several sentences/sections were removed from the first 4 paragraphs in accordance to the other reviewer's suggestions. Modified or new sections are marked in blue.
Reviewer 2 Report
This manuscript contains an extensive description of several ubiquitin shuttle proteins and receptors and of their function within the UPS and autophagy. Paragrahs 5, 6 and 7 are particularly novel, as they compare several interesting findings and put them in a context. In this view I would recommend the authors to keep the first part (paragraphs 1 to 4) to a minimum. This will help the reader to stay focused. Many details on the UPS and the ubiquitin machinery, the ubiquitin code, or on some of the shuttles could be omitted and other recent reviews can be mentioned instead. On the other hand, the autophagy molecular mechanism should be at least cited and the function of at least Atg8, including its similarity to ubiquitin, should be described.
Other general comments:
-in more than occasion, such as in in lines 221-227 the authors mention that there are two publication that obtain opposite conclusions. I think it might really be helpful to the reader if, rather than just mentioning the fact, the authors would try to suggest a possible explanation for it.
-When initially introducing Figure 2, it should be made clear to the reader which criteria have been used to select these proteins. Is it because they are all scientifically proven “shuttles” ? It will also help to include references in this figure or in the legend
-Many redundancies can be avoided throughout the text . For example (but it is just one of many samples) in line 252 “by the proteasome shuttle factor, Ubiquilin2 “ there is no need to explain what Ubiquilin 2 is, since it has been already mentioned earlier in the text.
-In many cases, such as in lines lines 42-51. there are terminology imprecisions (compare this part to the cited review by Dikic, 2017). I strongly recommend the author to correct the terminology wherever necessary.
-each scientific conclusion should be followed by the appropriate reference. Many references are missing or not cited when necessary.
Author Response
We addressed all reviewer suggestions. To improve the clarity of the text and keep the reader's focus on the main scope of this manuscript several sentences/sections were removed from the first 4 paragraphs. Modified or new sections are marked in blue.
Q: This manuscript contains an extensive description of several ubiquitin shuttle proteins and receptors and of their function within the UPS and autophagy. Paragrahs 5, 6 and 7 are particularly novel, as they compare several interesting findings and put them in a context. In this view I would recommend the authors to keep the first part (paragraphs 1 to 4) to a minimum. This will help the reader to stay focused. Many details on the UPS and the ubiquitin machinery, the ubiquitin code, or on some of the shuttles could be omitted and other recent reviews can be mentioned instead. On the other hand, the autophagy molecular mechanism should be at least cited and the function of at least Atg8, including its similarity to ubiquitin, should be described.
A: Acording to the reviewer's suggestion paragraphs 1-4 were shortened. Details about the UPS and its machinery were removed and proper review articles were cited instead.
Additionally more information was added about the autophagy and the Atg8 protein (including its similarity to ubiquitin). Proper litterature was cited.
Other general comments:
Q:-in more than occasion, such as in in lines 221-227 the authors mention that there are two publication that obtain opposite conclusions. I think it might really be helpful to the reader if, rather than just mentioning the fact, the authors would try to suggest a possible explanation for it.
A: In each instance of such case possible explanation of obtained contradictory conclusions was added.
Q:-When initially introducing Figure 2, it should be made clear to the reader which criteria have been used to select these proteins. Is it because they are all scientifically proven “shuttles” ? It will also help to include references in this figure or in the legend
A: To clarify this section the text was modified in accordance with the reviewer's suggestion. The legend was rewritten and references were included.
Q:-Many redundancies can be avoided throughout the text . For example (but it is just one of many samples) in line 252 “by the proteasome shuttle factor, Ubiquilin2 “ there is no need to explain what Ubiquilin 2 is, since it has been already mentioned earlier in the text.
A: Redundancies were removed.
Q:-In many cases, such as in lines lines 42-51. there are terminology imprecisions (compare this part to the cited review by Dikic, 2017). I strongly recommend the author to correct the terminology wherever necessary.
A: Terminology was corrected. The section containing lines 42-51 was removed to keep the initial paragraphs length to a minimum. Proper reviews were mentioned instead.
Q:-each scientific conclusion should be followed by the appropriate reference. Many references are missing or not cited when necessary.
A: Many references were added and cited whenever it seemed necessary.
Reviewer 3 Report
The authors have presented an informative review of the role of ubiquitin binding proteins and other regulatory mechanisms controlling the degradative fate of ubiqitinated proteins, whether through proteasomal or autophagic processes.
I found the review to be well organized and it provides good background, for the proposed model, with suitable references throughout. The description of the relevant evidence for the model is very good.
I have only one major concern. Although the paper is generally well-written and readable, there are a number of sentences that need grammatical or content revision for good readability. For instance:
Line 139 May help to give the alternate name (p62) for SQSTM1 in its first usage since many non-experts may be more familiar with it.
Lines 147-148 Sentence needs revision
Line 198 Substitute another descriptor to replace "professional" scaffold protein".
Lines 331-333 Sentence is hard to read, needs revision.
Lines 408-409 Sentence needs revision.
Lines 411-414 Sentence needs revision.
Lines 514-515 Sentence needs revision.
Lines 686 - 690 Section of Fig. 3 legend needs revision.
Line 688 "DUB" isn't found in the figure. I would either add it to the figure or remove from legend.
Lines 717-718 Sentence needs revision
Author Response
To improve the clarity of the text and to keep the reader's focus on the main scope of this manuscript, several sentences/sections were removed from the first 4 paragraphs in accordance to the other reviewer's suggestions. Modified or new sections are marked in blue.
Q: I have only one major concern. Although the paper is generally well-written and readable, there are a number of sentences that need grammatical or content revision for good readability. For instance:
Line 139 May help to give the alternate name (p62) for SQSTM1 in its first usage since many non-experts may be more familiar with it.
A: In the first sentence mentioning SQSTM1 an alternate name, p62, was also included.
Q: Lines 147-148 Sentence needs revision
Lines 331-333 Sentence is hard to read, needs revision.
Lines 408-409 Sentence needs revision.
Lines 411-414 Sentence needs revision.
Lines 514-515 Sentence needs revision.
Lines 717-718 Sentence needs revision
A: All sentences mentioned above were revised and corrected. The whole text was additionally checked for clarity and some sentences were rewritten as a result.
Q: Line 198 Substitute another descriptor to replace "professional" scaffold protein".
A: The sentence was corrected. The word "professional" was removed.
Q: Lines 686 - 690 Section of Fig. 3 legend needs revision.
Line 688 "DUB" isn't found in the figure. I would either add it to the figure or remove from legend.
A: The legend was revised and rewritten. Figure 3 was modified to include "DUB" as well as "E1-E2-E3 cascade".
Round 2
Reviewer 2 Report
This new version of this manuscript has fixed many problems that were present in the first version and now provides a clear and exhaustive description of ubiquitin-binding protein shuttles and their role in autophagy and protein degradation.